# CALIBFREE: Self-Supervised Feature Disentanglement for Calibration-Free Multi-Camera Multi-Object Tracking

## Abstract

Multi-camera multi-object tracking (MCMOT) faces significant challenges in maintaining consistent object identities across varying camera perspectives, particularly when precise calibration and extensive annotations are required. In this paper, we present CALIBFREE, a self-supervised representation learning framework that does not need any calibration or manual labeling for the MCMOT task. By disentangling view-agnostic and view-specific features through single-view distillation and cross-view reconstruction, our method adapts to complex, dynamic scenarios with minimal overhead. Experiments on the MMP-MvMHAT dataset show a 3% improvement in overall accuracy and a 7. 5% increase in the average F1 score over state-of-the-art approaches, confirming the effectiveness of our calibration-free design. Moreover, on the more diverse MvMHAT dataset, our approach demonstrates superior over-time tracking and strong cross-view performance, highlighting its adaptability to a wide range of camera configurations.

## 1 Introduction

Multiple Object Tracking (MOT) is an essential problem in computer vision, aiming to identify and track multiple objects within video streams. While single-camera tracking has been extensively studied Cao et al. (2023); Wojke et al. (2017); Cai et al. (2022); Meinhardt et al. (2022); Zeng et al. (2022); Zhang et al. (2023), the importance of Multi-Camera Multi-Object Tracking (MCMOT) continues to grow with the rising applications of multi-camera systems in surveillance, smart cities, and autonomous vehicles Gilbert & Bowden (2006); He et al. (2020); You & Jiang (2020); Cheng et al. (2023); Zhang et al. (2022a); Gu et al. (2023). MCMOT aims to maintain consistent object identities across multiple camera views, addressing inherent challenges such as viewpoint variation, occlusions, and synchronization issues, as illustrated in Figure 1. By integrating diverse viewpoints, MCMOT can offer improved tracking robustness, enhanced scene understanding, and fewer blind spots compared to single-camera methods Han et al. (2020; 2021).

Despite these advantages, achieving effective MCMOT remains challenging He et al. (2020); You & Jiang (2020); Chen et al. (2014). A primary difficulty arises from significant variations in object appearance and motion across different camera views, making reliable object re-identification (ReID) nontrivial. Moreover, many MCMOT methods Ristani et al. (2016b); Maksai et al. (2017); Tesfaye et al. (2019); He et al. (2020); You & Jiang (2020); Cheng et al. (2023) rely on calibrated camera setups or large-scale annotations. Even minor camera shifts—such as relocating a camera or changing its angle—can break calibration, causing immediate performance declines until the system is recalibrated and annotated data are recollected. Similarly, transitioning to a new scene often necessitates gathering a fresh dataset, performing calibration, and retraining the model. As camera networks expand or reconfigure, the associated computational overhead grows, making frequent recalibration and reannotation both costly and impractical in real-world applications.

**Main Results:** To address these limitations, we propose a self-supervised learning framework specifically designed for multi-camera setups with overlapping fields of view. Our method avoids explicit calibration and reduces the need for annotations by leveraging data-driven representation learning. In particular, we present a disentangled feature learning strategy that separates view-agnostic and view-specific features through single-view distillation and cross-view reconstruction.

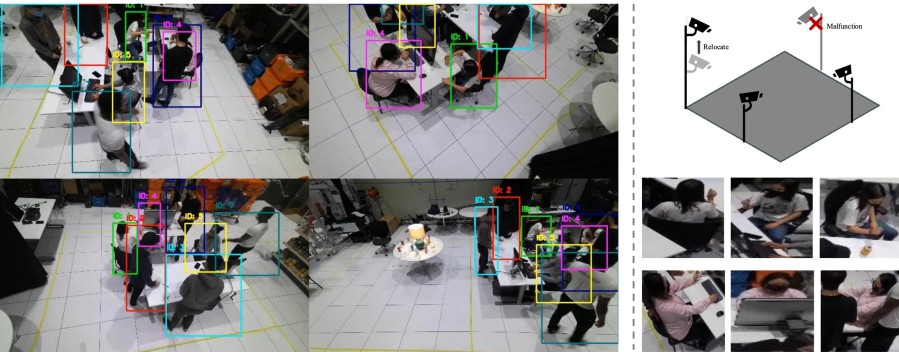

Figure 1: **Multi-Camera Multi-Object Tracking (MCMOT) setup.** *Left:* Multi-view scenes with individuals tracked across overlapping camera views, each assigned a unique color-coded bounding box. *Top right:* Flexible camera configurations illustrating variable camera numbers and placements across scenarios, or due to factors like malfunction or relocation. *Bottom right:* Examples of appearance variations for individuals across different viewpoints, highlighting the challenge of maintaining consistent identity association in multi-view tracking.

This approach mitigates viewpoint-based discrepancies and improves cross-view tracking without costly manual calibration or any labeling. Our contributions can be summarized as follows:

1. We propose a self-supervised representation learning framework for MCMOT, effectively reducing reliance on both manual annotations and camera calibration.

2. We introduce a disentangled feature learning strategy via single-view distillation and cross-view reconstruction, enhancing robustness against viewpoint variations.

3. We empirically validate our method on two challenging MCMOT datasets: (1) MMP-MvMHAT, featuring densely placed indoor cameras that capture crowded, occluded environments, where our method surpasses state-of-the-art baselines by 3% in overall accuracy and 7.5% in average F1 score. (2) MvMHAT, containing both indoor and outdoor scenes with sparser camera coverage and reduced overlapping fields of view, where our approach likewise demonstrates superior over-time tracking and strong cross-view performance, underscoring its adaptability to diverse real-world scenarios.

## 2 RELATED WORK

### 2.1 SINGLE-CAMERA MULTI-OBJECT TRACKING

Single-camera multi-object tracking (MOT) has been extensively studied, with the tracking-by-detection paradigm being the most widely adopted Cao et al. (2023); Leal-Taixé et al. (2016); Schulter et al. (2017); Wojke et al. (2017). In this framework, object detectors Duan et al. (2019); Girshick (2015); Ge et al. (2021) identify objects in each frame, and temporal associations are made using methods like the Kalman Filter Welch (1995) and the Hungarian Matching algorithm Kuhn (1955). Deep appearance features further improve association accuracy Chu & Ling (2019); Xu et al. (2019; 2020). End-to-end approaches such as MOTR Zeng et al. (2022), MOTRv2 Zhang et al. (2023), and TrackFormer Meinhardt et al. (2022) leverage query-based object detection to perform long-term tracking without manual association rules Carion et al. (2020). TransTrack Sun et al. (2020) and P3Aformer Zhao et al. (2022) improve efficiency using location-based cost matrices. However, single-camera MOT struggles with occlusions and complex interactions due to limited viewpoints, motivating research in multi-camera multi-object tracking (MCMOT).

### 2.2 MULTI-CAMERA MULTI-OBJECT TRACKING

Multi-camera multi-object tracking (MCMOT) has gained growing attention for its complexity and broad applications. Existing methods fall into three main categories: distributed, global, and end-to-end. *Distributed methods* perform tracking independently within each camera, followed by cross-view association Gilbert & Bowden (2006); Prosser et al. (2008); Cai & Medioni (2014); Chen et al.

(2014). Techniques such as hierarchical clustering Murtagh & Contreras (2012) and non-negative matrix factorization (NMF) Wang & Zhang (2012) are used to merge intra-camera tracklets, though they often assume ideal conditions not met in dynamic environments. *Global methods* detect individuals across views and associate them directly to build tracklets Ristani et al. (2016b); Maksai et al. (2017); Tesfaye et al. (2019). TRACTA He et al. (2020) and DMCT You & Jiang (2020) utilize perspective models and occupancy heatmaps, while ReST Cheng et al. (2023) employs a reconfigurable graph for robust association. *End-to-end methods* like MUTR3D Zhang et al. (2022a), PF-Track Pang et al. (2023), and ViP3D Gu et al. (2023) are designed for 3D tracking tasks such as autonomous driving. MCTR Niculescu-Mizil et al. (2024) proposes a calibration-free framework using track embeddings, but it depends on labeled data and fixed camera setups, limiting flexibility. Our work focuses on overlapping-camera scenarios, where shared fields of view offer appearance and geometric constraints for trajectory linking. In contrast, non-overlapping setups Javed et al. (2005); Tesfaye et al. (2017); Chilgunde et al. (2004) face challenges such as time delays and lack of spatial correspondence.

### 2.3 Self-Supervised Learning

Self-supervised learning (SSL) has become a powerful paradigm in computer vision Doersch et al. (2015); Noroozi & Favaro (2016) and multi-modal tasks Akbari et al. (2021); Wang et al. (2021), enabling robust representation learning without labeled data. Pretext tasks like context prediction Doersch et al. (2015); Pathak et al. (2016), jigsaw puzzles Noroozi & Favaro (2016); Kim et al. (2018b), and colorization Larsson et al. (2017) have shown effectiveness for image-level learning. For videos, pace prediction Wang et al. (2020) and space-time cube puzzles Kim et al. (2018a) help capture temporal dynamics. More recent techniques such as contrastive learning Chen et al. (2020); He et al. (2019); van den Oord et al. (2018) and masked autoencoding He et al. (2021); Huang et al. (2022) are effective across both image and video domains. In multi-object tracking, self-supervised methods use spatial-temporal consistency to reinforce object identity. Strategies include cross-input consistency Bastani et al. (2021), cycle-consistency Yin et al. (2023), and path-consistency Lu et al. (2024). In MCMOT, recent approaches like MvMHAT Feng et al. (2024) and MvMHAT++ Gan et al. (2021) leverage consistency-based tasks such as symmetric-consistency (SymC) and transitive-consistency (TrsC), though they rely on CNN-based features. In contrast, our method adopts masked autoencoding He et al. (2021), which is more compatible with transformer architectures and enables richer, more adaptable representation learning in complex MCMOT settings.

## 3 Method

In this section, we present the details of our proposed approach, **CALIBFREE**. We begin by formulating the problem, followed by a description of our algorithm, and conclude with how the generated features are used during inference.

### 3.1 Problem Formulation

Multi-camera multi-object tracking (MCMOT) aims to track all subjects across synchronized video streams from $V$ cameras and associate identities across views. This can be formulated as a spatio-temporal association problem with two objectives:

- **Intra-camera tracking:** Given detections $D_t^v = \{D_i^v \mid i = 1, 2, \ldots, N_t^v\}$ at frame $t$ in view $v$, associate them over time to form tracklets $\tau_t^v$, as in single-camera MOT.

- **Cross-view matching:** Match detections $\bar{D}_t = \{\bar{D}_t^1, \bar{D}_t^2, \ldots, \bar{D}_t^V\}$ across views at time $t$ that belong to the same subject.

Like single-camera methods (e.g., DeepSORT Wojke et al. (2017), ByteTrack Zhang et al. (2022b)), MCMOT relies on robust feature representations to ensure reliable association. These features must remain consistent across time and camera viewpoints while being discriminative enough to separate different identities.

Given all detections at time $t$, $D_t = \{D_{t,1}^1, \ldots, D_{t,N_t^V}^V\}$, the goal is to extract two types of features for each detection $D_{t,i}^v$:

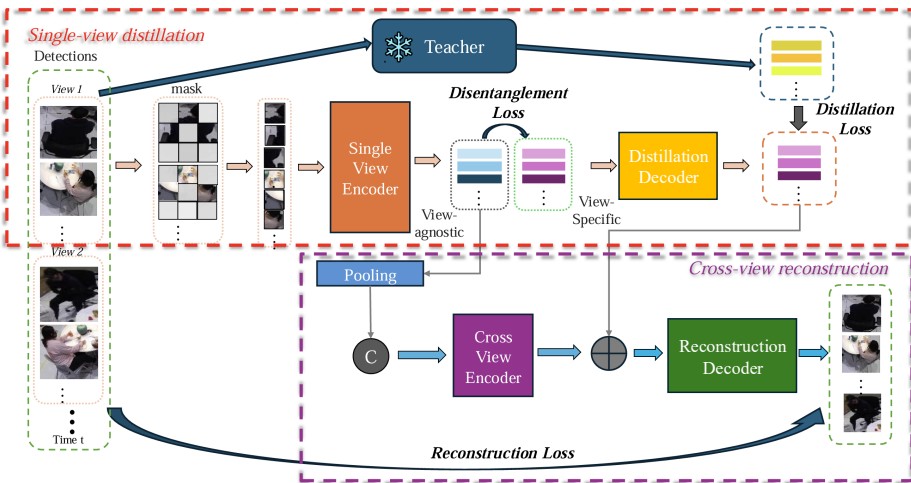

Figure 2: **Overview of CALIBFREE.** The method includes single-view distillation, feature disentanglement, and cross-view reconstruction. In single-view distillation (red box), masked detections are encoded, and feature quality is supervised by a teacher model using distillation loss. The disentanglement module splits features into view-agnostic and view-specific parts. For cross-view reconstruction (purple box), pooled view-agnostic features are processed to reconstruct masked patches across views, optimized with reconstruction loss.

- **View-agnostic features** ($f_a$): Capture identity-preserving cues (e.g., silhouette, body shape, pose) for cross-view matching.

- **View-specific features** ($f_s$): Encode appearance-specific details (e.g., clothing, texture) useful for temporal tracking within a view.

These features support both within-view and cross-view association, enabling robust identity continuity across space and time in uncalibrated multi-camera environments.

## 3.2 CALIBFREE

Masked autoencoders He et al. (2021) have proven effective in learning visual semantics, generating high-quality features from images. CALIBFREE builds on this framework to improve representations for detections of different persons, see Figure 2. Unlike traditional methods that reconstruct from partial observations within the same image and view, CALIBFREE introduces a cross-view reconstruction task, enabling reconstruction from observations across different views using view-agnostic features. Furthermore, it incorporates a distillation process from large models to refine the learning of view-specific features.

**Pre-processing.** At each timestep $t$, $V$ frames are captured from $V$ cameras. An off-the-shelf detector is applied to each frame to generate bounding boxes for all visible persons. The detected regions are cropped and resized to a uniform size $(H, W)$. Since the number of detections can vary across views, the maximum number of detections $N$ is used as a preset. For views with fewer detections, zero tensors of size $(H, W, C)$ are added to represent missing detections. The resulting input is $D_t = \{D_{t,i}^j \in \mathbb{R}^{V \times N \times H \times W \times C} \mid i = 1, 2, \ldots, N; j = 1, 2, \ldots, V\}$, which consolidates all detections from all views at time $t$.

**Masking.** Each detection is divided into non-overlapping patches, $P = \{P_i \mid P_i \in \mathbb{R}^{C \times h \times w}\}_{i=1}^M$, where $M = \frac{H}{h} \times \frac{W}{w}$ is the total number of patches. These patches are converted into a sequence of tokens, $K = \{K_i \mid K_i \in \mathbb{R}^E\}_{i=1}^M$, using patch embedding and positional encoding. A subset of tokens $K^{vis} \subset K$ (e.g., 25%) is randomly sampled without replacement, and the remaining tokens are masked, following a "random masking" strategy. Although various masking strategies exist, Random masking is chosen for its simplicity and ease of implementation without requiring additional inputs.

The same mask is applied across all detections $D_t$ to ensure consistency between views. This shared mask preserves positional encoding and prevents disruptions in cross-view reconstruction, which relies on consistent masking across views, as discussed later.

**Single View Encoder.** The single-view encoder $\Phi_{sve}$ is a standard Vision Transformer (ViT Dosovitskiy et al. (2021)) applied to the $M^{vis}$ visible, unmasked patch tokens $K^{vis} \subset K$. Unlike conventional masked autoencoders, the encoder processes all unmasked tokens from detections within each view, enabling multi-head self-attention across patches in a single view. This setup captures variations between different detections, with consistent masked token positions enhancing cross-detection learning.

Positional embeddings for the patch tokens are generated using a sinusoidal function across all detection patches within a view. This ensures that while unmasked tokens may occupy the same positions across detections because of consistent mask, their positional embeddings remain distinct.

The encoder outputs features split evenly into view-agnostic features $f_a$ (first half) and view-specific features $f_s$ (second half), as follows: $f_a, f_s = \Phi_{sve}(K^{vis}), \quad f_a, f_s \in \mathbb{R}^{M^{vis} \times \frac{E}{2}}$

**Distillation Decoder.** We project the view-specific encoder features to the decoder width $E_d$ with a linear layer and concatenate learned embeddings for the masked positions to form a length-$M$ token sequence. This sequence is fed to a shallow ViT decoder $\Phi_{\text{distill}}$. Positional embeddings are added to *all* tokens so that masked tokens retain their spatial coordinates.

The decoder outputs per-patch features for the entire detection, $\hat{f}_s \in \mathbb{R}^{M \times E_d}$. In parallel, the corresponding unmasked crop is processed by a pretrained teacher to obtain patch-level targets. Before computing the distillation loss, a linear head is used to align the student features $f_{student}$ to the teacher feature space $f_{teacher}$.

We use the publicly released ViT-L MAE model He et al. (2021) pretrained on ImageNet-1K (self-supervised). Its single-view pretraining emphasizes view-specific cues—e.g., color, fine textures, and local details—while contributing less to view-agnostic properties such as aspect ratio, coarse silhouette, or pose. This makes it a good supervisor for the view-specific branch (via patch-level distillation) while leaving the cross-view branch to learn identity-consistent signals across cameras.

**Cross View Encoder.** The view-agnostic features are passed through a pooling layer to combine patch information, producing a single view-agnostic embedding per detection. Note that no information is mixed across cameras at this stage—only patches within the same detection are combined. All embeddings from each view are then projected into the cross-view encoder dimension, $E_d$, and sent to a shallow ViT-based cross-view encoder. Multi-head self-attention is applied across these embeddings to capture differences between views. The output feature $\hat{f}_a \in \mathbb{R}^{E_d}$ is learnt through all the views, representing the high-level semantic features that are universal across views.

**Reconstruction Decoder.** The view-agnostic feature $\hat{f}_a$ is combined with the view-specific feature $\hat{f}_s$ for each patch, creating an enriched representation that captures both cross-view consistency and camera-specific details. These combined features are fed into the reconstruction decoder, which reconstructs the original image by predicting pixel values for each masked patch.

During decoding, each output vector from the decoder represents the pixel values of a specific patch, effectively reconstructing masked areas. The decoder's final layer uses a linear projection to match the total pixel count per patch, preserving each patch's spatial structure. After projection, the output is reshaped to form a coherent, reconstructed image, closely resembling the original input.

**Loss.** To ensure robust training, we employ a combination of three losses:

*Disentanglement Loss*: This normalized mutual information (NMI) loss measures the independence between view-agnostic and view-specific features, quantifying how much information about one feature set is shared by the other:

$$L_{\text{disentangle}} = NMI(f_a, f_s)$$

Minimizing $L_{\text{disentangle}}$ enhances feature disentanglement by reducing shared information between the two feature sets.

*Distillation Loss*: This loss facilitates knowledge transfer from a larger teacher model pretrained on a different dataset. Given potential domain differences, Smooth L1 Loss is used to mitigate the

impact of outliers:

$$L_{\text{distillation}} = \text{SmoothL1}(f_{\text{student}}, f_{\text{teacher}})$$

*Reconstruction Loss*: This loss calculates the mean squared error (MSE) between the reconstructed and original images in pixel space, applied only to masked patches:

$$L_{\text{reconstruction}} = \text{MSELoss}(f_{\text{reconstructed}}^{\text{masked}}, f_{\text{original}}^{\text{masked}})$$

The overall loss function combines these components:

$$\text{Loss} = L_{\text{disentangle}} + L_{\text{distillation}} + L_{\text{reconstruction}}$$

### 3.3 INFERENCE

A key advantage of **CALIBFREE** is its independence from camera calibration and human annotations. While both the single-view encoder/decoder and the cross-view encoder are used during self-supervised training, only the single-view encoder is needed at inference.

During inference, all patches are passed (unmasked) through the single-view encoder to generate feature embeddings. These features are average-pooled across patches to produce a single embedding per detection, which is then split into view-agnostic and view-specific components. For single-camera tracking, we integrate the view-specific features into DeepSORT Wojke et al. (2017) for within-camera association, using Kalman filtering to refine tracks. For cross-camera matching, we use the view-agnostic features to compute the association matrix, without applying any Kalman filter.

## 4 RESULTS

Due to page limits, we include dataset, evaluation metrics, implementation details and more results in the Appendix.

### 4.1 MAIN RESULTS AND ANALYSIS

#### 4.1.1 BASELINE METHODS

We compare our method against state-of-the-art approaches in Tables 1 and 2, where the best results are highlighted and second-best underlined.

For single-camera (over-time) tracking, we include four representative MOT methods: Tracktor++ Bergmann et al. (2019), CenterTrack Zhou et al. (2020), TraDeS Wu et al. (2021), and TrackFormer Meinhardt et al. (2022). Since these do not support cross-view tracking, we assign ground-truth IDs upon first appearance in each camera and apply each method independently within views.

For MCMOT, we evaluate DeepCC Ristani & Tomasi (2018) and SVT Dong et al. (2021), with DeepCC leveraging an off-the-shelf ReID model Zhong et al. (2017) for cross-view association. We also include MvMHAT and its extension MvMHAT++, two self-supervised methods that require no fine-tuning, with MvMHAT++ introducing an additional training stage. All MCMOT methods are evaluated on the MMP-MvMHAT dataset using ground-truth bounding boxes for consistency, except that previous self-supervised methods are also tested with YOLOX-generated detections.

Detector-based results using YOLOv7 and YOLOX are shown in Table 4. For MvMHAT, we use a Detectron2 Wu et al. (2019) detector; among available models, we select the ResNet-50 variant that achieves MOTA closest to the original MvMHAT paper for fair comparison.

Lastly, we do not re-train supervised baselines on our dataset, as they require labeled data—unlike our self-supervised approach—ensuring a fair comparison in terms of generalization and calibration-free capability.

#### 4.1.2 RESULTS ON MMP-MVMHAT

**Over-Time Tracking:** Table 1 reports CALIBFREE's performance on the indoor-focused MMP-MvMHAT dataset, where many subjects exhibit limited motion (e.g., seated office workers). Despite

| Methods | Over-time Tracking | | | | | Cross-View Tracking | | | | Overall | |
|---|---|---|---|---|---|---|---|---|---|---|---|
| | IDP | IDR | IDF1 | MOTA | HOTA | AIDP | AIDR | AIDF1 | MHAA | A | F |
| *supervised* | | | | | | | | | | | |
| Tracktor++Bergmann et al. (2019) | 67 | 56 | 61 | 67.2 | 46.2 | 62 | 23.2 | 33.8 | 19.1 | 43.1 | 47.4 |
| CenterTrackZhou et al. (2020) | 35.9 | 24.1 | 28.8 | 48.2 | 27.1 | 29.3 | 3.9 | 6.8 | 3.1 | 25.7 | 17.8 |
| TraDeSWu et al. (2021) | 59.7 | 50.1 | 54.5 | 66.1 | 42.7 | 54.5 | 17.3 | 26.2 | 13.3 | 39.7 | 40.4 |
| TrackFormerMeinhardt et al. (2022) | 41.8 | 28.6 | 34 | 46.7 | 30.2 | 39.9 | 5.3 | 9.4 | 3.2 | 25 | 21.7 |
| DeepCCRistani & Tomasi (2018) | 51.6 | 52.5 | 52.1 | 92.5 | 59.3 | 42.7 | 23.4 | 30.2 | 19.7 | 56.1 | 41.2 |
| SVTDong et al. (2021) | 63.1 | 63.4 | 63.3 | 96.7 | 68.8 | 53.8 | 33.4 | 41.2 | 29.8 | 63.1 | 52.3 |
| *self-supervised* | | | | | | | | | | | |
| MvMHAT(YOLOX)Gan et al. (2021) | 51.1 | 53.2 | 52.7 | 82.1 | 47.2 | 30.4 | 17.1 | 23.2 | 14.1 | 48.1 | 37.9 |
| MvMHAT(GT)Gan et al. (2021) | 58.6 | 58.8 | 58.7 | 93.7 | 65 | 35.4 | 21.2 | 26.5 | 20.3 | 57 | 42.6 |
| MvMHAT++(YOLOX)Feng et al. (2024) | 59.3 | 60.5 | 60.1 | 82.2 | 52.1 | 45.7 | 34.1 | 40.5 | 28.9 | 55.5 | 50.3 |
| MvMHAT++(GT)Feng et al. (2024) | 67.1 | 67.6 | 67.3 | 95 | 70.2 | 62.1 | 42.5 | 50.4 | 40.6 | 67.8 | 58.9 |
| **CALIBFREE(YOLOX)** | 81.3 | 77.1 | 79.1 | 82.5 | 59.2 | 52.3 | 48.4 | 50.3 | 34.4 | 58.4 | 64.7 |
| **CALIBFREE(GT)** | 82.2 | 78 | 80 | 97.6 | 75.7 | 55 | 50.9 | 52.8 | 44.2 | 70.8 | 66.4 |

Table 1: **Results on MMP-MvMHAT.** CALIBFREE surpasses both supervised and self-supervised methods across key metrics, demonstrating robust identity tracking in over-time and cross-view scenarios.

| Methods | Over-time Tracking | | | | | Cross-View Tracking | | | | Overall | |
|---|---|---|---|---|---|---|---|---|---|---|---|
| | IDP | IDR | IDF1 | MOTA | HOTA | AIDP | AIDR | AIDF1 | MHAA | A | F |
| *supervised* | | | | | | | | | | | |
| Tracktor++Bergmann et al. (2019) | 54.2 | 40.1 | 46.1 | 66.5 | 42.8 | 34.3 | 14.6 | 20.5 | 37.1 | 51.8 | 33.3 |
| CenterTrackZhou et al. (2020) | 44.3 | 33.5 | 38.1 | 63.5 | 37.8 | 29.7 | 9.1 | 13.9 | 34.1 | 48.8 | 26.0 |
| TraDeSWu et al. (2021) | 46.7 | 43.2 | 44.9 | 69.5 | 47.3 | 32.4 | 14.0 | 19.6 | 36.0 | 52.8 | 32.2 |
| TrackFormerMeinhardt et al. (2022) | 52.3 | 47.2 | 49.6 | 70.4 | 47.3 | 47.8 | 23.2 | 31.3 | 40.2 | 55.3 | 40.4 |
| DeepCCRistani & Tomasi (2018) | 44.7 | 44.2 | 44.4 | 63.9 | 41.1 | 57.9 | 34.8 | 43.4 | 43.8 | 53.9 | 43.9 |
| SVTDong et al. (2021) | 47.9 | 47.2 | 47.6 | 65.4 | 43.1 | 61.7 | 45.7 | 52.5 | 50.4 | 56.9 | 50.0 |
| *self-supervised* | | | | | | | | | | | |
| MvMHATGan et al. (2021) | 53.1 | 52.0 | 52.5 | 64.7 | 47.9 | 53.0 | 46.4 | 49.5 | 51.7 | 58.2 | 51.0 |
| MvMHAT++Feng et al. (2024) | 58.5 | 57.4 | 57.9 | 66.3 | 51.8 | 63.8 | 56.0 | 59.6 | 59.7 | 63.0 | 58.8 |
| **CALIBFREE** | 59.1 | 58.4 | 58.7 | 60.4 | 52.0 | 58.9 | 56.2 | 57.4 | 57.0 | 58.4 | 58.1 |

Table 2: **Results on MvMHAT.** CALIBFREE achieves best or second best results across most key metrics.

the simplicity of such motion—often inflating IDF1 for other methods—CALIBFREE achieves an IDF1 of 80.0, demonstrating strong identity continuity under occlusion and crowding. Its HOTA score of 75.7 reflects balanced accuracy in detection and association, minimizing ID switches and ensuring stable long-term tracking.

**Cross-View Tracking:** CALIBFREE achieves an AIDF1 of 52.8 and MHAA of 44.2, outperforming all baselines. While its AIDP is slightly lower than MvMHAT++, CALIBFREE achieves higher AIDR, indicating better recall of cross-camera matches. This reflects its ability to capture identity-consistent features under challenging viewpoint changes and appearance similarity. Compared to supervised baselines like DeepCC and SVT—which require manual annotations—CALIBFREE delivers stronger association without labels. Center-based trackers (e.g., CenterTrack) lack robust appearance modeling and accumulate ID switches in crowded scenes, while TrackFormer can produce mismatches when its detection step underperforms.

**Sensitivity to Bounding Box Quality:** As shown in Table 1, CALIBFREE is more robust to noisy bounding boxes than prior self-supervised methods using YOLOX detections. While others suffer significant performance drops, CALIBFREE maintains ID-related metrics with minimal degradation, highlighting the resilience of its learned features to imperfect detections.

**Overall:** CALIBFREE surpasses self-supervised baselines (MvMHAT, MvMHAT++) in both accuracy (70.8) and F1 score (66.4), demonstrating the advantage of its disentangled features for both temporal and spatial consistency. Although motion in MMP-MvMHAT is simpler, the cluttered indoor layout introduces frequent identity ambiguities, which CALIBFREE handles effectively.

### 4.1.3 RESULTS ON MvMHAT

**Over-Time Tracking.** Table 2 presents results on the MvMHAT dataset, which includes both indoor and outdoor scenes with sparsely placed cameras and minimal overlap. Single-camera methods like TrackFormer yield competitive MOTA (e.g., 70.4), largely reflecting detector quality. However, CALIBFREE achieves higher ID-based metrics (IDP, IDR, IDF1) and HOTA, indicating better temporal identity consistency.

**Cross-View Tracking.** Single-view trackers struggle when ID switches occur, propagating errors across views and degrading AIDF1. That said, their cross-view performance improves slightly over MMP-MvMHAT due to reduced overlap and fewer direct transitions. CALIBFREE outperforms multi-view trackers like DeepCC and SVT across all cross-view metrics. MvMHAT++ achieves

| Methods | Over-time Tracking | | | | | Cross-View Tracking | | | | Overall | |
|---|---|---|---|---|---|---|---|---|---|---|---|
| | IDP | IDR | IDF1 | MOTA | HOTA | AIDP | AIDR | AIDF1 | MHAA | A | F |
| ViT-L (teacher) | 82.1 | 77.8 | 79.9 | 97.5 | 75.6 | 47.9 | 44.4 | 46.1 | 36.7 | 67.1 | 63 |
| Distillation only | 78.4 | 67.8 | 72.7 | 97.2 | 68.4 | 47.5 | 40.1 | 43.5 | 34.4 | 65.8 | 58.1 |
| Reconstruction only | 81.2 | 77.1 | 79.1 | 97.5 | 75.2 | 52.3 | 48.4 | 50.3 | 41.8 | 69.7 | 64.7 |
| **CALIBFREE(ours)** | **82.2** | **78** | **80** | **97.6** | **75.7** | **55** | **50.9** | **52.8** | **44.2** | **70.8** | **66.4** |

Table 3: **Ablation studies of CALIBFREE variations.** The full model, combining distillation, reconstruction, and feature disentanglement, achieves the best performance across all tracking metrics.

| Detector Pretrain | Detector Inference | Over-time Tracking | | | | | Cross-View Tracking | | | | Overall | |
|---|---|---|---|---|---|---|---|---|---|---|---|---|
| | | IDP | IDR | IDF1 | MOTA | HOTA | AIDP | AIDR | AIDF1 | MHAA | A | F |
| YOLO7 | YOLO7 | 59.9 | 57.5 | 58.6 | 44.9 | 47.7 | 47.2 | 44.3 | 45.7 | 16.4 | 30.7 | 52.2 |
| Groundtruth | YOLO7 | 59.8 | 57.4 | 58.5 | 44.9 | 47.7 | 49.3 | 46.1 | 47.6 | 18.3 | 31.6 | 53.1 |
| YOLOX | YOLOX | 81.3 | 77.1 | 79.1 | 82.5 | 59.2 | 52.3 | 48.4 | 50.3 | 34.4 | 58.4 | 64.7 |
| Groundtruth | YOLOX | 81 | 76.8 | 78.9 | 82.5 | 59 | 54 | 50.1 | 52 | 37.5 | 60 | 65.5 |
| **Groundtruth** | **Groundtruth** | **82.2** | **78** | **80** | **97.6** | **75.7** | **55** | **50.9** | **52.8** | **44.2** | **70.8** | **66.4** |

Table 4: **Ablation study on detector choice during pretraining and inference.** Results show that CALIBFREE maintains high ID association consistency, with inference detector choice impacting tracking accuracy more than pretraining.

higher precision and AIDF1 in this setting, which we attribute to two factors: (1) CALIBFREE uses a less accurate detector (lower MOTA), reducing cross-view consistency; and (2) MvMHAT++ benefits from a second-stage training step specifically tailored for cross-view association, offering an advantage in sparsely overlapped environments.

**Overall Performance.** CALIBFREE demonstrates strong gains over most single-camera and multi-view baselines in overall accuracy ($A$) and F1 score ($F$). By disentangling view-specific and view-agnostic features, it maintains identity across views without calibration. Although MvMHAT++ excels in some cross-view metrics, CALIBFREE's unified, annotation-free framework delivers robust and generalizable performance under real-world challenges like occlusions, sparse views, and subject similarity.

## 4.2 ABLATION STUDIES

### 4.2.1 EFFECT OF DISTILLATION AND RECONSTRUCTION

Table 3 presents an ablation study comparing four CALIBFREE variants. The *ViT-L (teacher)* setting uses features from a pretrained ViT-L Dosovitskiy et al. (2021) directly, without training a student. *Distillation only* trains a student using only the distillation loss $\mathcal{L}_{distill}$, while *Reconstruction only* trains a student from scratch using only $\mathcal{L}_{recon}$. Our *full model* combines $\mathcal{L}_{distill} + \mathcal{L}_{recon} + \mathcal{L}_{disent}$ and outputs disentangled view-specific and view-agnostic features.

The ViT-L teacher achieves strong over-time tracking (IDF1: 79.9, MOTA: 97.5) but limited cross-view performance (AIDF1: 46.1, MHAA: 36.7). Distillation alone underperforms due to reduced model capacity and masked inputs (AIDF1: 43.5). Reconstruction alone slightly lowers over-time performance (IDF1: 79.1) but improves cross-view accuracy (AIDF1: 50.3), highlighting the importance of spatial reconstruction for multi-view consistency. The full CALIBFREE model achieves the best overall results (F: 66.4, Accuracy: 70.8), validating the importance of all components for robust uncalibrated tracking.

### 4.2.2 EFFECT OF DETECTOR CHOICE

Table 4 compares three detector configurations—ground truth, YOLOX Ge et al. (2021), and YOLOv7 Wang et al. (2023)—used during both pretraining and inference. CALIBFREE demonstrates strong robustness to detector choice during pretraining; however, inference quality has a more substantial effect. Switching from YOLOX to the less accurate YOLOv7 results in noticeable performance drops, highlighting the importance of reliable detections—a challenge common to all tracking methods. Notably, models pretrained with ground truth and inferred using YOLOX achieve ID-based metrics comparable to those with ground-truth inference, demonstrating CALIBFREE's adaptability when the inference detector maintains reasonable accuracy. Moreover, as shown in Table 1, CALIBFREE is significantly more resilient to bounding box imperfections compared to previous self-supervised methods.

| Teacher Model | Student Model | Over-time Tracking | | | | | Cross-View Tracking | | | | Overall | |
|---|---|---|---|---|---|---|---|---|---|---|---|---|
| | | IDP | IDR | IDF1 | MOTA | HOTA | AIDP | AIDR | AIDF1 | MHAA | A | F |
| **ViT-L** | **ViT-B** | **82.2** | **78** | **80** | **97.6** | **75.7** | **55** | **50.9** | **52.8** | **44.2** | **70.8** | **66.4** |
| ViT-B | ViT-B | 82.0 | 77.8 | 79.8 | 97.5 | 75.5 | 52.1 | 47.8 | 49.7 | 42.0 | 69.8 | 64.8 |
| ViT-B | ViT-S | 77.6 | 75.8 | 76.7 | 97.2 | 71.8 | 50.3 | 45.4 | 47.7 | 40.4 | 68.8 | 62.2 |

Table 5: **Ablation studies of different models.** Using ViT-L as teacher and ViT-B as student achieves best results.

| Mask ratio | Over-time Tracking | | | | | Cross-View Tracking | | | | Overall | |
|---|---|---|---|---|---|---|---|---|---|---|---|
| | IDP | IDR | IDF1 | MOTA | HOTA | AIDP | AIDR | AIDF1 | MHAA | A | F |
| 0.9 | **82.3** | 78 | **80.1** | 97.6 | **75.8** | 53.8 | 48.7 | 51.1 | 43.0 | 70.3 | 65.6 |
| **0.75** | 82.2 | **78** | 80 | **97.6** | 75.7 | 55 | **50.9** | **52.8** | **44.2** | **70.8** | **66.4** |
| 0.5 | 82.0 | 77.9 | 79.8 | 97.5 | 75.6 | **55.1** | 50.1 | 52.4 | 43.9 | 70.7 | 66.1 |

Table 6: **Ablation studies of mask ratios.** 0.75 achieves the best balance between over-time and cross-view tracking.

### 4.2.3 IMPACT OF TEACHER AND STUDENT MODEL SIZES

Table 5 investigates three teacher–student setups: ViT-L→ViT-B, ViT-B→ViT-B, and ViT-B→ViT-S. A larger teacher (ViT-L) improves cross-view performance (AIDF1: 49.7→52.8, MHAA: 42.0→44.2) due to richer representations feeding into the cross-view encoder. Over-time metrics (IDF1, MOTA) remain mostly unchanged, suggesting that temporal continuity is less sensitive to teacher size. On the other hand, reducing the student to ViT-S lowers both over-time (IDF1: 76.7) and cross-view (AIDF1: 47.7) performance, indicating insufficient capacity for robust identity modeling. Although smaller students are more efficient, this comes at the cost of accuracy—especially in complex multi-view scenarios.

### 4.2.4 EFFECT OF MASK RATIOS IN PRETRAINING

We examine the impact of mask ratios (0.5, 0.75, 0.9) in Table 6. High masking (e.g., 0.9) harms cross-view performance (AIDF1: 51.1, MHAA: 43.0), suggesting that excessive masking limits the model's ability to learn spatially consistent features. Over-time metrics (IDF1, MOTA) remain stable, as temporal tracking depends less on detailed spatial information. While both 0.5 and 0.75 achieve comparable cross-view results (AIDF1: 52.4 vs. 52.8), the 0.75 setting reduces token usage, offering better efficiency. Thus, a 0.75 mask ratio strikes the best balance between accuracy and computational cost for both temporal and cross-view tracking.

## 5 CONCLUSION, LIMITATION, AND FUTURE WORK

We have introduced **CALIBFREE**, a self-supervised multi-camera multi-object tracking (MCMOT) method that achieves state-of-the-art performance without relying on camera calibration or manual annotations. By disentangling view-agnostic and view-specific features, supported by cross-view reconstruction and knowledge distillation, CALIBFREE robustly handles complex identity associations across time and views. Experiments on the MMP-MvMHAT and MvMHAT datasets underscore its strong adaptability in both over-time and cross-view tracking.

Despite these advances, our approach remains limited by its exclusive use of RGB features, thereby overlooking valuable geometric relationships across views. Although learning geometric associations without camera parameters is nontrivial, incorporating such information could enhance identity consistency and cross-view associations. In future work, we aim to explore self-supervised methods for integrating geometric cues, enabling CALIBFREE to better leverage spatial relationships between detections.

**Ethics Statement.** This work uses publicly available datasets (MvMHAT, MMP-MvMHAT) with no personally identifiable information or human-subject interaction. We adhere to the ICLR Code of Ethics. While multi-camera tracking may raise privacy concerns, our contributions are intended for academic research and do not enable identification beyond provided annotations.

**Reproducibility Statement.** Model architecture, objectives, training schedules, and evaluation protocols are specified in Secs. 3–4 and the appendix; ablations (Sec. 4.4) support design choices.

Datasets and preprocessing steps are documented, and we will release code, pretrained weights, and evaluation scripts to reproduce all reported results upon publication.

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

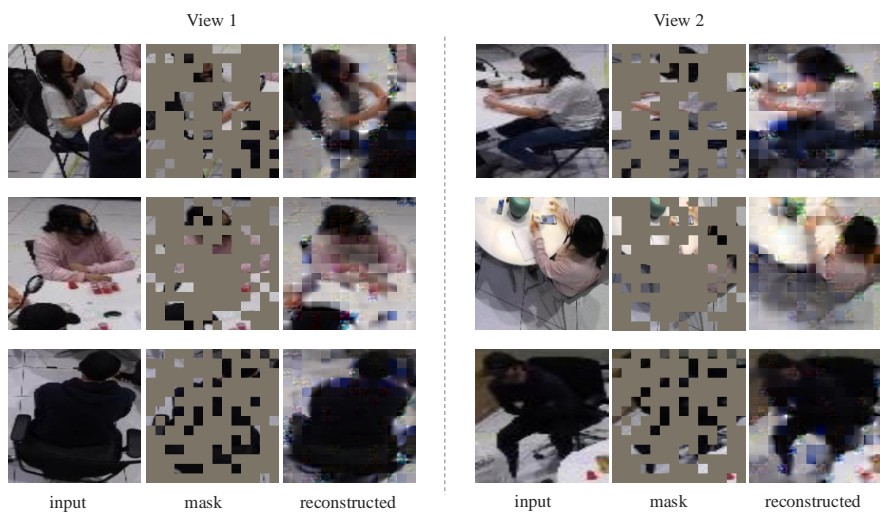

Figure 3: **Cross-view reconstruction results.** The input images are the original scenes, while the masked images indicate regions removed for reconstruction. CALIBFREE effectively reconstructs these masked regions, capturing identity-preserving details across viewpoints even when large portions are obscured.

## A  KEY TAKEAWAYS AND CALIBRATION INDEPENDENCE

**Calibration-free design.** Our approach never accesses camera intrinsics, extrinsics, or homographies at any stage of training or inference. As a result, deployment to a new camera network requires no calibration effort, and the method remains robust even if cameras are re-positioned or experience drift over time (Section. 1, Figure. 1).

**Disentangled feature learning.** A central insight is that explicitly disentangling two complementary embeddings—learned via a masked auto-encoder without any camera metadata—is both feasible and beneficial. The *view-specific* branch preserves appearance nuances tied to a particular camera, while the *view-agnostic* branch captures identity cues consistent across viewpoints. This separation (i) integrates seamlessly with off-the-shelf trackers such as DeepSORT Wojke et al. (2017), (ii) eliminates calibration and labeling overhead, and (iii) opens new directions for cross-source representation learning (e.g., audio–video or LiDAR–camera).

## B  DATASETS

**MMP-MvMHAT.** Adapted from MMPTRACK mmp, MMP-MvMHAT features 4–6 overlapping indoor cameras and 28 individuals. It provides 8,000 fully annotated frames across four training scenes and 4,000 frames for validation, with no calibration data. This setup, focused on crowded and occluded environments, poses a challenging multi-view tracking task.

**MvMHAT.** MvMHAT Feng et al. (2024) is a large-scale dataset containing 26 video groups (98 sequences) sourced from Campus Xu et al. (2016), EPFL Fleuret et al. (2008), and newly collected footage. Each group includes 3–4 synchronized camera views, totaling over 90,000 annotated frames. Split into training and testing (13 groups each) with a 2:1 ratio, MvMHAT covers diverse scenarios and camera angles—often with 90° viewpoint differences—to facilitate robust multi-view tracking evaluation.

## C  EVALUATION METRICS

**Over-time Tracking.** We adopt Multiple Object Tracking Accuracy (MOTA) Bernardin & Stiefelhagen (2008) to assess single-view tracking performance in terms of false positives, missed detections, and identity switches. Given the emphasis on robust identity association over time, we further use ID Precision (IDP), ID Recall (IDR), and ID F1 (IDF1) Ristani et al. (2016a), as well as

High Order Tracking Accuracy (HOTA) Luiten et al. (2021) for a balanced evaluation of detection, association, and localization.

**Cross-view Tracking.** For multi-camera scenarios, we use Association ID Precision (AIDP), Association ID Recall (AIDR), and Association ID F1 (AIDF1) Han et al. (2020; 2021), which average pairwise matching accuracy across different cameras. We also include Multi-view Multi-Human Association Accuracy (MHAA), which penalizes identity-consistency errors in multi-camera contexts with frequent occlusions and appearance shifts.

**Overall.** To provide a holistic MCMOT assessment, we calculate the MCMOT F1 score ($F$) and accuracy score ($A$) by taking the average of F1 and accuracy across both over-time and cross-view tracking Feng et al. (2024):

$$F = \text{Mean(IDF1, AIDF1)} \quad , \quad A = \text{Mean(MOTA, MHAA)}.$$

## D  IMPLEMENTATION DETAILS

**Pretraining Phase.** After detecting the bounding box for each person, the region of interest (ROI) is cropped based on the bounding box coordinates and resized to a fixed resolution of $224 \times 224$ pixels through upsampling or downsampling. These resized ROIs are divided into non-overlapping patches of size $16 \times 16$.

The *single-view encoder* is implemented as a vanilla Vision Transformer (ViT) base model with 12 transformer blocks and 12 attention heads, using an embedding dimension of 768. During inference, the output of the single-view encoder is split into view-agnostic and view-specific features, each with a dimension of 384.

The *single-view decoder*, *cross-view encoder*, and *cross-view decoder* are shallow Vision Transformer models, each comprising 8 attention blocks and 16 attention heads with an embedding dimension of 512. For knowledge distillation, the *teacher model* is a ViT large model pretrained on ImageNet, with an output feature dimension of 1024. To align dimensions during distillation, the 512-dimensional output of the single-view decoder is projected to 1024 dimensions using a linear layer.

Pretraining is conducted over 400 epochs with a base learning rate of $1.5 \times 10^{-4}$ and a 40-epoch warmup phase. The weight decay is set to 0.05. Reconstruction loss is computed using mean squared error (MSE) loss applied to normalized pixel values rather than raw pixel values. The maximum number of detections is set to be 10 for both datasets.

**Inference Phase.** During inference, a detector identifies bounding boxes for all persons, and the ROIs are cropped based on the bounding box coordinates. Features for each detection are generated by feeding all patches (without masking) to the single-view encoder. The per-patch features are then aggregated using max pooling to produce a single feature vector (matching the encoder dimension) representing each detection.

For *over-time tracking*, we utilize DeepSort, with the generated features serving as the primary matching criterion, complemented by a Kalman filter as a secondary criterion. For *cross-view tracking*, aggressive matching is employed, computing pairwise similarity between detections from different views to establish associations.

## E  VISUALIZATION

Figure 3 illustrates how CALIBFREE captures identity-preserving features across different viewpoints, showing the original input, masked patches, and reconstructed outputs. Substantial portions of each subject are masked to simulate partial observations, yet CALIBFREE reliably restores these regions by leveraging both view-agnostic and view-specific features. Crucial details like posture, clothing texture, and overall silhouette remain largely intact, supporting consistent identity tracking across camera views. Even when significant information is obscured, the model reconstructs occluded areas accurately based on the available visible patches, all without requiring camera calibration. This visualization underscores CALIBFREE's robustness and practical utility in multi-camera scenarios with frequent occlusions and substantial viewpoint variations.

# F COMPUTATION AND RUNTIME

We report the hardware setup and training time for each dataset:

| Dataset | #GPUs | GPU Model | Mem/GPU | Batch | Epochs | Wall-clock |
|---|---|---|---|---|---|---|
| MMP-MvMHAT | 4 | NVIDIA H100 | 80 GB | 5 | 400 | 13.4 h |
| MvMHAT | 4 | NVIDIA A100 | 80 GB | 5 | 400 | 38.3 h |

At inference, using four cameras with 1080P input resolution, a ViT-B encoder (as single-view encoder) for each view, and DeepSORT as the tracker, the average runtime per timestep is 154 ms (± 32 ms) on one NVIDIA A100 80GB GPU.

# G USE OF LARGE LANGUAGE MODELS (LLMS).

We used LLMs only as a writing assistant for improving clarity and conciseness of our text (e.g., rewriting and rephrasing). LLMs were not involved in research ideation, design, experimentation, or analysis. The authors take full responsibility for all content.

