# OpenReview forum: "CALIBFREE: Self-Supervised Feature Disentanglement for Calibration-Free Multi-Camera Multi-Object Tracking"
_ICLR.cc/2026/Conference — ICLR 2026 Conference Withdrawn Submission_

### Official Review · Reviewer_v86L · 2025-10-28

**Soundness:** 2
**Presentation:** 1
**Contribution:** 2
**Rating:** 2
**Confidence:** 5

**Summary:**

This paper introduces CALIBFREE, a self-supervised representation learning framework for Multi-Camera Multi-Object Tracking (MCMOT). The core innovation is a feature disentanglement strategy—using single-view distillation and cross-view reconstruction—designed to separate view-agnostic identity features ($\mathbf{f}_a$) from view-specific appearance features ($\mathbf{f}_s$), thereby eliminating the need for camera calibration and manual labeling. CALIBFREE claims state-of-the-art performance on the MMP-MvMHAT and MvMHAT datasets.

**Strengths:**

- CALIBFREE tackles the significant practical challenge in MCMOT concerning the dependence on precise camera calibration and large-scale annotations
- The approach shows strong performance gains, achieving a 3% improvement in overall accuracy and a 7.5% increase in F1 score over SOTA self-supervised baselines on the challenging MMP-MvMHAT dataset

**Weaknesses:**

- The evaluation is strictly limited to two specialized multi-camera tracking benchmarks: MMP-MvMHAT (indoor, crowded, dense camera overlap) and MvMHAT (indoor/outdoor, sparse camera coverage). While these datasets are challenging for cross-view feature learning, the paper provides no evidence of the method's generalization to standard or conventional Multi-Object Tracking (MOT) or Multi-Camera Multi-Object Tracking (MCMOT) benchmarks
- For the over-time tracking results reported in Tables 1 and 2, a significant portion of the comparison involves single-camera MOT methods (e.g., Tracktor++, CenterTrack, TraDeS, TrackFormer). These methods are inherently disadvantaged in the MCMOT setup because their cross-view tracking performance is measured artificially by assigning ground-truth IDs upon first appearance in each camera
- The inference procedure (Section 3.3) is over-simplified as it omits critical details regarding identity management and conflict resolution, which are essential for robust performance in a distributed Multi-Camera Multi-Object Tracking (MCMOT) framework. Although CALIBFREE utilizes view-specific features for single-camera DeepSORT tracking and view-agnostic features for "aggressive matching" across views, the methodology for deciding which identity to inherit or how to resolve contradictions between the local tracklets and cross-view associations is not explicitly defined

**Questions:**

1. Following the third bullet point in Weakness section, I have two major questions about the inferencing step:
1-1. The Inference section (Sec. 3.3) describes a two-stage process involving intra-camera DeepSORT tracking (using $f_s$) followed by cross-view "aggressive matching" (using $f_a$). Please clarify the precise mechanism, model, or optimization strategy used to resolve identity conflicts when local tracklets established by DeepSORT contradict the global associations proposed by the pairwise similarity matrix derived from $f_a$. Specifically, what rules or algorithms (e.g., hierarchical clustering, Non-Negative Matrix Factorization (NMF)) are employed to assign the final global ID and ensure track consistency in cases of track merging?
1-2. Can the authors provide implementation details regarding the initialization and termination of tracklets within the DeepSORT component (e.g., frame thresholds for track confirmation/deletion)? Furthermore, given the focus on robust tracking in crowded and occluded scenarios, does CALIBFREE maintain a centralized, long-term tracklet bank or trajectory memory for re-association across extended time delays (beyond the immediate DeepSORT time window), especially in scenarios involving sparse camera coverage, such as the MvMHAT dataset?
2. The evaluation is confined to MMP-MvMHAT and MvMHAT; does CALIBFREE demonstrate its strong generalization capability on conventional MOT benchmarks (e.g., MOT17, MOT20) or standard calibrated MCMOT datasets, and how does its performance compare if labels/calibration were available?
3. CALIBFREE's cross-view feature disentanglement approach is validated primarily on the specialized MMP-MvMHAT and MvMHAT datasets, relying exclusively on RGB appearance features while admitting that "valuable geometric relationships across views" are overlooked. Given that DIVOTrack [A] focuses on Multi-Camera Multi-Object Tracking (MCMOT) in "Diverse Open Scenes" which often require robust spatial constraints, have the authors investigated how CALIBFREE's performance compares to the DIVOTrack baseline on highly diverse or geometrically complex outdoor scenes, particularly in its current state without integrating geometric cues?
4. The paper discusses the importance of the normalized mutual information (NMI) Disentanglement Loss ($L_{disentangle} = NMI(f_a, f_s)$). Could the authors provide a more detailed analysis or visualization demonstrating the qualitative difference between the view-agnostic features ($f_a$) and view-specific features ($f_s$)?
[A} DIVOTrack: A Novel Dataset and Baseline Method for Cross-View Multi-Object Tracking in DIVerse Open Scenes
, Hao et al., https://arxiv.org/abs/2302.07676

---

### Official Review · Reviewer_7a4W · 2025-11-01

**Soundness:** 1
**Presentation:** 1
**Contribution:** 2
**Rating:** 2
**Confidence:** 4

**Summary:**

this paper investigates multi-camera multi-object tracking (MCMOT) and follows previous method to take a tracking-by-detection approach. specifically, it introduces a so-called "self-supervised" method for the bounding box feature representations, which is claimed to benefit from "cross view reconstruction". on the MMP-MvMHAT dataset, the proposed method achieves relatively good performance.

**Strengths:**

+ cross-view reconstruction for multi-camera settings makes sense

**Weaknesses:**

- very unclear methodology. the technical ambiguities are severe enough that even after careful inspection, the reviewer still cannot understand the proposed method.
  - how many patch tokens are fed into the Single View Encoder for one forward pass? what is the number $M^{vis}$ for the visible tokens in L220? is it a quater of the tokens in one dection bbox $\frac{M}{4}$ (suggested by the similar named visible token set)? or is it the summation of all $N$ detection bboxes' patches within a view $N\times \frac{M}{4}$ (Fig. 3 and L222 "Unlike conventional masked autoencoders, the encoder processes all unmasked tokens from detections within each view, enabling multi-head self-attention across patches in a single view")?
  - what is the Cross View Encoder? what is its input? there is no proper decription either in text or in Fig. 3. is it all $V\times N\times \frac{M}{4}$ patches from all $V$ views, $N$ detection per-view?
  - what is the reconstruction target in Reconstruction Decoder? again, no proper details here. is it the same view? is it different views only? or is it both? also, why only detection bboxes from view 1 are included in Fig. 3? this is a must-fix issue for a paper claiming "cross-view reconstruction" as its core contribution
  - how to ensure $f_a$ is really 'view-agnostic'? what supervision?
- why the proposed method can be called "self-supervised"? it seems that proper cross-view annotations on human bboxes are needed to establish the same identity, which the reviewer believes is a strong supervision signal.
- no discussion or comparison with very related work [r1]
- no ablation study
- both Fig. 1 and the Introduction section emphasize the flexibility towards camera layout. yet, no experiment on flexible camera layout (location & rotation adjustments, random shut down) are provided to support these claims.
- poor readability & notation. please use \citep for citation in parentheses (similar citation issues occur throughout the entire manuscript). also, for math notations, using the capital cases for both sets (e.g., $K$ for patches) and numbers (e.g., $M$ for the number of patches) can cause a lot of confusion.

[r1]. Weinzaepfel, Philippe, Vincent Leroy, Thomas Lucas, Romain Brégier, Yohann Cabon, Vaibhav Arora, Leonid Antsfeld, Boris Chidlovskii, Gabriela Csurka, and Jérôme Revaud. "Croco: Self-supervised pre-training for 3d vision tasks by cross-view completion." Advances in Neural Information Processing Systems 35 (2022): 3502-3516.

**Questions:**

see above

---

### Official Review · Reviewer_pfm2 · 2025-11-02

**Soundness:** 3
**Presentation:** 3
**Contribution:** 3
**Rating:** 8
**Confidence:** 3

**Summary:**

The paper proposes a self-supervised representation learning framework for calibration-free multi-view multi-object tracking (MCMOT). The method learns view-agnostic and view-specific features through a combination of single-view distillation and cross-view reconstruction, eliminating the need for camera intrinsics, extrinsics, or identity labels.
During training, each detected instance is divided into non-overlapping patches, and a shared random mask is applied across all synchronized views. The visible patches are processed by a single-view encoder that outputs disentangled view-agnostic and view-specific features. The view-specific branch is supervised by a teacher distillation decoder that reconstructs fine-grained patch features from a pretrained vision transformer. The cross-view encoder, implemented as a multi-head self-attention module, aggregates object-level embeddings from all views to produce a shared, view-consistent latent representation. A reconstruction decoder then combines the cross-view and distillation outputs to reconstruct the original views, enforcing alignment across cameras. The overall loss combines three terms: (1) normalized mutual information loss for feature disentanglement, (2) distillation loss for spatial detail preservation, and (3) masked reconstruction loss for cross-view consistency. The  method is evaluated on MvMHAT and MMP-MvMHAT, and overall performs better than both supervised and self-supervised baselines.

**Strengths:**

1. The problem is well motivated. Manual track annotation is time consuming and sensor extrinsics can change over time. The paper utilizes the idea of masked autoencoders to learn view-agnostic and view-specific features without annotation and calibration information.
2. The proposed method performs better than supervised methods for temporal and cross-view tracking on MMP-MvMHAT and MvMHAT. Although for MvMHAT dataset, it performs lower than MvMHAT++.

**Weaknesses:**

1. The is calibration free but requires synchronized frames. Furthermore, the calibration-free assumption doesn’t seem to be any different from MvMHAT or MvMHAT++.
2. All evaluations are done using static camera setup. There are no experiments with dynamic cameras which would show the strength of the calibration-free claim.
3. Inference requires A100 and runs at ~154 ms which is significant runtime given the compute.
4. Minor: There are several duplicate citations. E.g., Ristani 2016a and Ristani 2016b. Yonatan 2017 and 2019.

**Questions:**

1. What specific mechanisms enforce separation between view-specific and view-agnostic features (beyond the NMI term)?
2. How exactly are the two feature sets combined in the reconstruction decoder?
3. How is the mutual information normalized (geometric mean, average, or max-entropy)?
4. How robust is the model to temporal desynchronization between views?
5. Can the method generalize to dynamic camera configurations without retraining?

---

### Official Review · Reviewer_sk1G · 2025-11-03

**Soundness:** 1
**Presentation:** 2
**Contribution:** 1
**Rating:** 2
**Confidence:** 5

**Summary:**

The paper presents a self-supervised representation learning framework that does not need any calibration or manual labeling for the MCMOT task. The paper proposes to separate view-agnostic and view-specific features through single-view distillation and cross-view reconstruction.

**Strengths:**

The paper writing is clear.

**Weaknesses:**

- The motivation of the paper appears weak. It is unclear MCMOT system specifically requires a self-supervised learning approach. This motivation does not directly stem from the challenges outlined in the introduction, such as significant appearance or motion variations across cameras or minor calibration shifts. Moreover, being calibration-free does not inherently require self-supervision, as numerous prior works [A, B] have addressed this issue through alternative means. The paper should clarify the necessity and unique benefit of self-supervised learning in this context, and discuss its advantages over existing methods.

- The problem definition lacks rigor. The relationship between D and \bar{D} is unclear. If detections in one scene can be cross-referenced across views, how are these sets differentiated or connected? This conceptual gap makes the formulation difficult to interpret.

- The methodology section lists several standard components: Masking, Distillation Decoder, Reconstruction Decoder, and Disentanglement Loss, without sufficiently explaining their specific roles or unique contributions. The rationale behind combining these elements should be explicitly discussed, especially how each part addresses the core challenges introduced earlier.

- The experimental validation is weak. The analysis primarily involves substituting off-the-shelf models, with limited evidence of how the proposed framework leads to measurable improvements. Stronger experimental design and ablation studies are needed to substantiate the claimed contributions.

- The paper claimed to address complex, dynamic scenarios, but only one dataset was reported.

[A] Quach, et al. Dyglip: A dynamic graph model with link prediction for accurate multi-camera multiple object tracking. CVPR 2021.

[B] Nguyen, et al. Lmgp: Lifted multicut meets geometry projections for multi-camera multi-object tracking. CVPR 2022.

**Questions:**

See weaknesses.

---

### Note · Authors · 2025-11-14

I have read and agree with the venue's withdrawal policy on behalf of myself and my co-authors.